# Computational design of novel nanobodies targeting the receptor binding domain of variants of concern of SARS-CoV-2

**Phoomintara Longsompurana[1], Thanyada Rungrotmongkol[2,3]\*, Nongluk Plongthongkum[1], Kittikhun Wangkanont[4,5], Peter Wolschann[6], Rungtiva P. Poo-arporn**[ID][1]\*

**1** Biological Engineering Program, Faculty of Engineering, King Mongkut's University of Technology Thonburi, Bangkok, Thailand, **2** Center of Excellence in Biocatalyst and Sustainable Biotechnology, Department of Biochemistry, Faculty of Science, Chulalongkorn University, Bangkok, Thailand, **3** Program in Bioinformatics and Computational Biology, Graduate School, Chulalongkorn University, Bangkok, Thailand, **4** Center of Excellence for Molecular Biology and Genomics of Shrimp, Department of Biochemistry, Faculty of Science, Chulalongkorn University, Bangkok, Thailand, **5** Center of Excellence for Molecular Crop, Department of Biochemistry, Faculty of Science, Chulalongkorn University, Bangkok, Thailand, **6** Institute of Theoretical Chemistry, University of Vienna, Vienna, Austria

\* thanyada.r@chula.ac.th (TR); rungtiva.pal@kmutt.ac.th (RPP)

**Data Availability Statement:** All relevant data for this study are within the paper, its Supporting information files, the OSF repository (https://osf.io/

## Abstract

The COVID-19 pandemic has created an urgent need for effective therapeutic and diagnostic strategies to manage the disease caused by the severe acute respiratory syndrome coronavirus 2 (SARS-CoV-2). However, the emergence of numerous variants of concern (VOCs) has made it challenging to develop targeted therapies that are broadly specific in neutralizing the virus. In this study, we aimed to develop neutralizing nanobodies (Nbs) using computational techniques that can effectively neutralize the receptor-binding domain (RBD) of SARS-CoV-2 VOCs. We evaluated the performance of different protein-protein docking programs and identified HDOCK as the most suitable program for Nb/RBD docking with high accuracy. Using this approach, we designed 14 novel Nbs with high binding affinity to the VOC RBDs. The Nbs were engineered with mutated amino acids that interacted with key amino acids of the RBDs, resulting in higher binding affinity than human angiotensin-converting enzyme 2 (ACE2) and other viral RBDs or haemagglutinins (HAs). The successful development of these Nbs demonstrates the potential of molecular modeling as a low-cost and time-efficient method for engineering effective Nbs against SARS-CoV-2. The engineered Nbs have the potential to be employed in RBD-neutralizing assays, facilitating the identification of novel treatment, prevention, and diagnostic strategies against SARS-CoV-2.

## Introduction

The emergence of COVID-19, caused by SARS-CoV-2, has resulted in a severe global public health emergency [1–3]. The infection process involves the receptor binding motif (RBM) of the SARS-CoV-2 RBD spike protein (S protein) attaching to human ACE2, a transmembrane protein receptor, resulting in virus entry through receptor-mediated endocytosis into the cell

fesr8), the figshare repository (https://figshare.com/s/070fed60428a4d6ca20b), and the Protocols.io repository (https://www.protocols.io/view/nb-design-for-sars-cov-2-rbd-cy93xz8n).

**Funding:** Research project is supported by the National Research Council of Thailand (NRCT) and King Mongkut's University of Technology Thonburi: N42A650316, the Research Strengthening Project of the Faculty of Engineering and Thailand Science Research and Innovation (TSRI), Basic Research Fund: Fiscal year 2023 (Program Smart Healthcare). P.L. gratefully acknowledge the financial support provided by the Petchra Pra Jom Klao Ph.D. Research Scholarship from King Mongkut's University of Technology Thonburi. T.R. is financially supported by the National Research Council of Thailand (NRCT, grant number N42A650231). The funders had no role in study design, data collection and analysis, decision to publish, or preparation of the manuscript.

**Competing interests:** The authors have declared that no competing interests exist.

[4]. Mutations in RBD, particularly in VOCs such as Alpha (B.1.1.7), Beta (B.1.351), Gamma (P.1), Delta (B.1.617.2), and Omicron (B.1.1.529); BA.1 and BA.2, have been shown to increase virus transmissibility, evade host defenses, and confer the ability to escape immune responses [4–6]. As a result, the development of high-specific neutralizing molecules such as nanobodies (Nbs) is critical to combat the emergence of VOCs and their impact on global public health [7–9].

Nbs, which are the single variable heavy chain domains of antibodies, are a promising option due to their small size, high binding affinity, solubility, stability, and manufacturability [10–12]. However, traditional methods for producing Nbs, such as phage-displayed libraries, are time-consuming, expensive, and require gene modification using genetic engineering techniques [10, 12–14]. To address these limitations, we propose an *in silico* approach to develop novel, high-specific, and broad neutralizing Nbs against SARS-CoV-2 RBDs. We evaluated seven protein-protein docking programs and identified the best program for Nb/RBD docking. Using HDOCK, we docked 29 Nbs against VOC RBDs to identify two lead Nbs, which were then engineered to create potentially broad-specific Nbs against VOC RBDs (Fig 1). The engineered Nbs are expected to promote a high affinity to Nb library, paving the way for next-generation Nb applications in COVID-19 diagnosis or therapeutics.

## Materials and methods

### Evaluation of the performance of docking programs

The protein data sets of 29 Nbs and 86 antibodies (Abs) in complex with RBDs for blind docking were retrieved from Protein Data Bank (PDB) (https://www.rcsb.org/) [15] (S1 Table) in the supplementary information. The RBD data sets included Wuhan-Hu-1 (Wh) and six VOC RBDs: Alpha, Beta, Delta, Gamma, Omicron BA.1 sub-lineage, and Omicron BA.2 sub-lineage (S2 Table). A list of ACE2 and other viral RBDs/HAs (S3 Table) was compiled to evaluate cross-binding. Before performing blind docking, we removed heteroatoms/molecules, such as metal ions, small molecules, water molecules, and His-tag, from all complexes, prepared the protein chains of RBDs and ligands (Nbs or antibodies) separately using Discovery Studio software, and retrieved the missing amino acids in the protein chain using the SWISS-MODEL expert system (https://swissmodel.expasy.org/) [16].

Blind docking of all RBDs and Nbs/antibodies was performed using seven protein-protein docking programs, including HDOCK (http://hdock.phys.hust.edu.cn/) [17–21], ATTRACT (www.attract.ph.tum.de.) [22], pyDockWEB (https://life.bsc.es/pid/pydockweb) [23], GRAMM-X (https://gramm.compbio.ku.edu/) [24], PatchDock (http://bioinfo3d.cs.tau.ac.il/PatchDock/) [25, 26], FRODOCK (http://frodock.chaconlab.org/) [27, 28], and ZDOCK (https://zdock.umassmed.edu/) [29]. The output of 100 docking poses was obtained from each docking program. The correlation between the rank pose (top or best pose) and native pose was calculated using backbone superimposition by CHIMERA software [30]. The root mean square deviation (RMSD) values of the ligands were calculated using the Discovery Studio program [31].

### Selection of lead Nbs

To identify the top two lead Nbs that exhibit broad and specific binding to the targeted RBDs, we conducted a selection process. This involved redocking all 29 Nbs listed in the S1 Table with the targeted RBDs using a blind docking method through the HDOCK. Subsequently, we minimized the energy of the Nb/RBD complexes utilizing the AMBER ff14SB force field, providing the complexes with the lowest binding energy. For each Nb and RBD, a separate

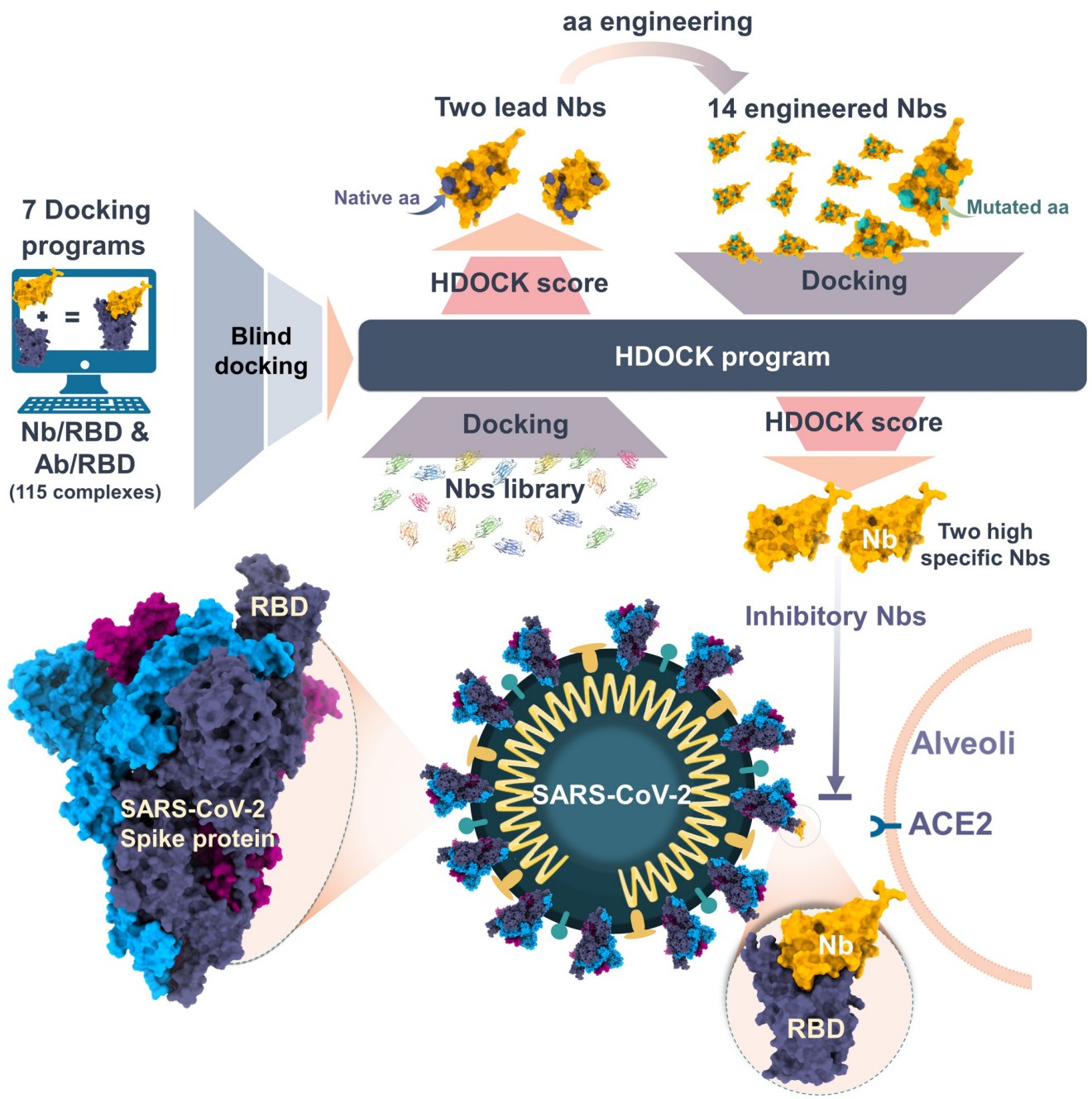

**Fig 1. An overview of the molecular docking procedure used for validating protein-protein docking and engineering nanobodies (Nbs) to develop broad-specific binding Nbs against VOC RBDs.**

redocking was performed using HDOCK to determine the best docking score. Following the blind docking procedure, we calculated RMSD values to assess the accuracy of the docking poses for the 29 Nbs with respect to all targeted RBDs. The similarity of amino acid sequences of 29 Nbs was analyzed using the Clustal Omega server (https://www.ebi.ac.uk/Tools/msa/clustalo/) [32]. The lead Nbs were selected based on the best mean HDOCK score, the lowest

mean RMSD, and their distinct amino acid sequences. These selected Nbs were then applied in the next step of the structure-based engineering procedure.

## Structural-based engineering and broad specific binding of Nbs

Prior to the procedure, the Nb/RBD complexes in native form, in their initial state after blind docking using the HDOCK, underwent an energy minimization process utilizing the AMBER ff14SB force field. This was done to align the torsion angles of complementary amino acid side chains between Nb and RBD, ensuring uniformity in the lowest binding energy and preventing any atom that might have astride. The HDOCK was used to perform redocking of the optimized Nb and RBD, yielding the native form's score as the initial score. To improve the binding affinity of Nbs to all targeted RBDs, the two lead Nbs were mutated using the site-direct mutagenesis feature on the Discovery Studio program. During the mutation process, Nb residues that had no interaction and repulsion with RBD were considered (S1 Fig). Prior to redocking, each mutated amino acid was subjected to optimization using the CHARM force field within the Discovery Studio program, resulting in the post-mutation score. After docking, the ΔHDOCK value was calculated for a single mutation of each Nb using Eq (1):

$$\Delta \text{HDOCK} = \text{HDOCK}_{\text{Nb (mutant)}} - \text{HDOCK}_{\text{Nb (native)}} \tag{1}$$

where ΔHDOCK represents the difference in HDOCK scores before and after a single mutation, $\text{HDOCK}_{\text{Nb (mutant)}}$ is the HDOCK score of the mutant Nb after mutation, and $\text{HDOCK}_{\text{Nb (native)}}$ is the HDOCK score of the native Nb before mutation. The HDOCK score is a scoring function that evaluates the binding affinity between the Nb and the RBD based on the intermolecular interactions. The mutated residue at a specific position that showed the lowest ΔHDOCK was chosen for multi-point mutation. The multi-mutated Nb sequences were aligned by BioEdit software for evaluating the sequence similarity. To investigate the broad specific binding of engineered Nbs, cross-docking between Nb and all targeted RBDs, ACE2, and other viral RBDs/HAs was performed using the HDOCK program.

## Physicochemical properties prediction of engineered Nbs

The predicted physicochemical properties of the engineered Nbs, including contact surface amino acid, chemical interaction, and their general physical properties, were determined using the PDBsum server (http://www.ebi.ac.uk/thornton-srv/databases/pdbsum/Generate.html) [33] to evaluate the intermolecular interaction between the engineered Nbs and targeted RBDs. Furthermore, the other physicochemical properties were predicted using the ProtParam (ExPASy) tool (https://web.expasy.org/protparam/) [34], the pI value was calculated using the Protein–Sol web server (https://protein-sol.manchester.ac.uk/) [35], and the total charge was calculated by PROTEIN CALCULATOR v3.4 (https://protcalc.sourceforge.net/) [36].

## Statistics

Statistical testing was conducted using SPSS software (IBM SPSS Statistics 25.0). Normality and homogeneity of variance were assessed with the Shapiro-Wilk test and Levene's test, respectively. Parametric or non-parametric tests were used for the analysis of variance. The performance of blind docking programs was evaluated with Kruskal-Wallis and Dunn's test. The cross-binding of designed Nbs with targeted RBDs or ACE2 was evaluated with one-way

ANOVA and Dunnett's test. The binding affinity of engineered Nb was compared to native Nb using an independent t-test.

## Results

### Evaluation of the performance of docking programs

The docking program's scoring function algorithm is a crucial factor that directly affects the accuracy of docking results. To evaluate the docking precision of seven selected docking programs, 115 ligand-RBD complex data sets were used, and 100 poses were generated from each program for each complex. These poses were ranked based on the scoring function and the RMSD from lowest to highest, and the first rank pose based on the scoring function of each program was termed "top pose", while the pose with the lowest RMSD among all the predictably generated poses was termed "best pose". However, the best docking program does not always generate the top pose and best pose at the same rank accordingly. Hence, we considered three primary criteria: the median of RMSD of 100 docking poses, the effect of the number of docking outputs, and the RMSD cut-off for evaluation.

For the first criterion, S2A, S2B and S3 Figs indicated that the HDOCK program demonstrated the lowest median RMSD of 0.75 Å and 0.70 Å for the top and best poses, respectively, which were lower than any other programs. The FRODOCK program also showed the second lowest median RMSD, 2.96 Å and 2.11 Å for the top and best poses, respectively. Although HDOCK and FRODOCK demonstrated excellent docking algorithms for top and best poses, HDOCK showed a smaller distribution than FRODOCK for the top and best poses. The second criterion (see S4 Fig), the effect of the number of rankings of docking output, was investigated by using the top 1, 25, 50, 75, and 100 pose(s) generated by different docking programs. From this analysis, the success rate was calculated as the percentage of the top poses that had RMSD cut-off $\leq$ 5.00 Å). From the results, HDOCK showed the lowest mean RMSD and highest success rate for all numbers of top docking poses, and its success rate was higher than 85% at the top 1 of posing rank. In contrast, FRODOCK showed a success rate of more than 80% for the top 25 to 100 poses, but their mean RMSD was higher than that of HDOCK.

Based on the RMSD cut-off (S5 Fig), the success rate of the top pose and best pose by HDOCK was higher than other programs at all RMSD cut-offs, especially for the low cut-off (RMSD $\leq$ 1.00 Å), which is considered as a highly accurate pose. Furthermore, HDOCK showed the highest success rate of 83.5% when the rank of the top pose was the same as the best pose (S2C Fig). The HDOCK also demonstrated success rates of 85% and 99% for the top and best pose, respectively (S6 Fig) when high accuracy, medium accuracy, and acceptable accuracy RMSD criteria were considered [31]. These results show that HDOCK's scoring function is suitable for the Nb/RBD or Ab/RBD systems [37–39]. Even though HDOCK can operate based on both template-base and template-free docking modes, we used template-free docking modes as blind or global docking. Thus, HDOCK's scoring function is reasonable for protein-protein docking [20]. Considering all the reasons above, HDOCK was selected for Nb/RBD docking in the next part of the Nb engineering process.

### Selection of lead Nbs

The selection of lead Nbs for broad-specific binding to RBDs was a challenging process. The criteria for selection included *(i)* the HDOCK score of the lead Nbs should be low, indicating high binding affinity, *(ii)* the correct poses of the lead Nbs on the target RBD should exhibit low RMSD values, *(iii)* their sequences should differ from each other while maintaining an

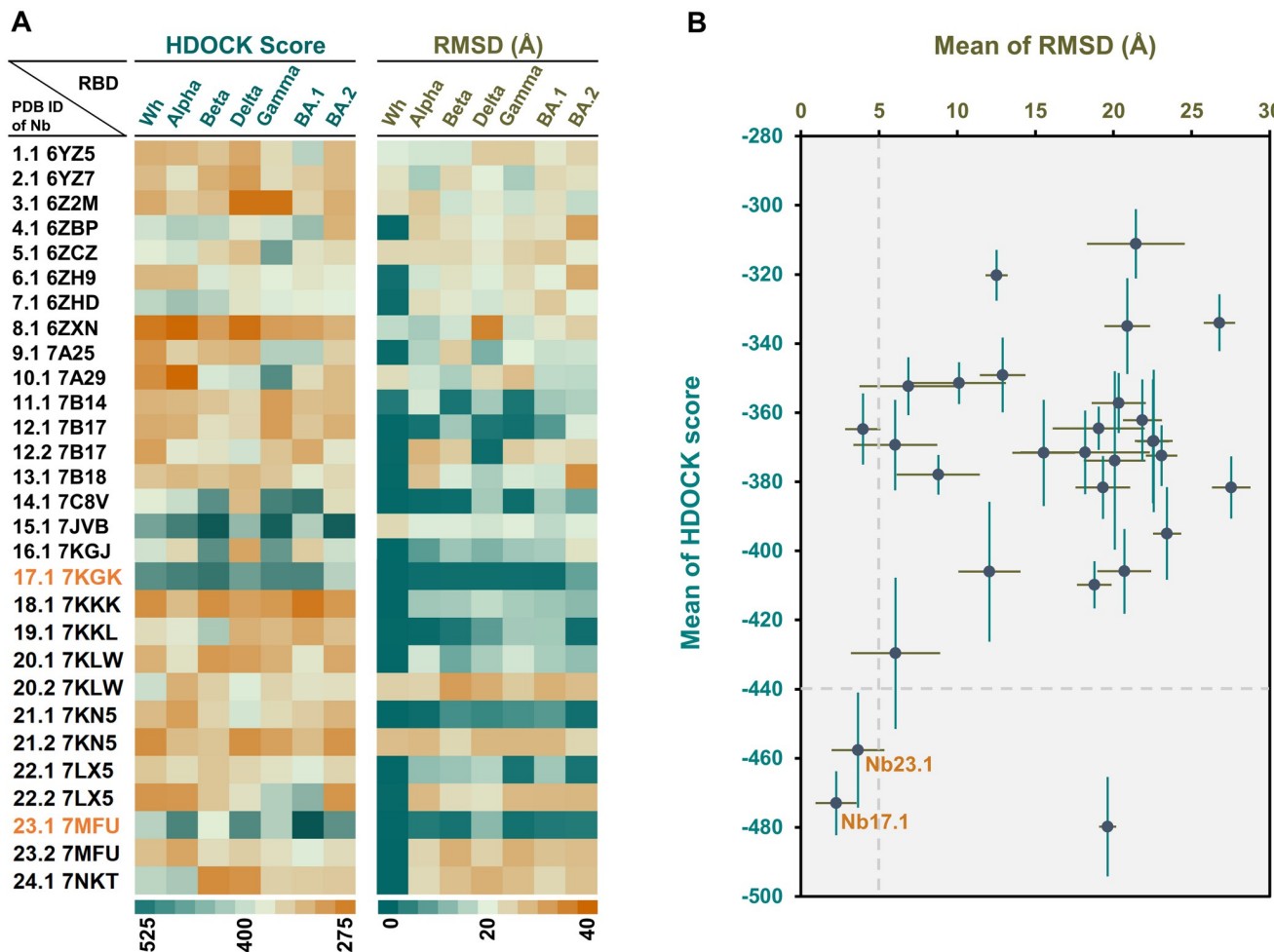

**Fig 2. Docking results of 29 Nbs on targeted RBDs.** (A) the heat map of HDOCK score and RMSD, (B) the plot of mean HDOCK score ± SEM *vs.*mean RMSD ± SEM (N = 7).

acceptable level of sequence similarity upon interaction with target RBDs, and *(iv)* the interaction sites of the lead Nbs on the receptor binding motif (RBM) of all RBDs should be distinct from each other. To select the lead Nbs, all 29 Nbs were docked against seven targeted RBDs using the HDOCK program. The results in Fig 2A and 2B show that Nb17.1 and Nb23.1 are the top two Nbs with the highest affinity to targeted RBDs, as indicated by their lowest mean HDOCK scores (-473.01 ± 8.99 for Nb17.1 and -457.63 ± 16.36 for Nb23.1) and lowest mean RMSD values (2.25 ± 1.25 Å for Nb17.1 and 3.65 ± 1.63 Å for Nb23.1), respectively. However, Nb15.1 had a low mean HDOCK score (-479.11 ± 14.07) but a high mean RMSD value (19.62 ± 0.48 Å) over the RMSD cut-off of 5.00 Å, suggesting that Nb15.1 is a non-specific binder to all target RBDs.

To further determine if Nb17.1 and Nb23.1 met all four selection criteria, the amino acid sequences at the interaction sites were compared. The results showed that Nb17.1 and Nb23.1 had 71.1% similarity (Fig 3A). Moreover, the structure and sequence alignment revealed that these two Nbs had different amino acid residues on the binding domain (Fig 3B). Finally, the docking poses of the two Nbs on target RBDs were observed. Fig 3C and 3D exhibit that

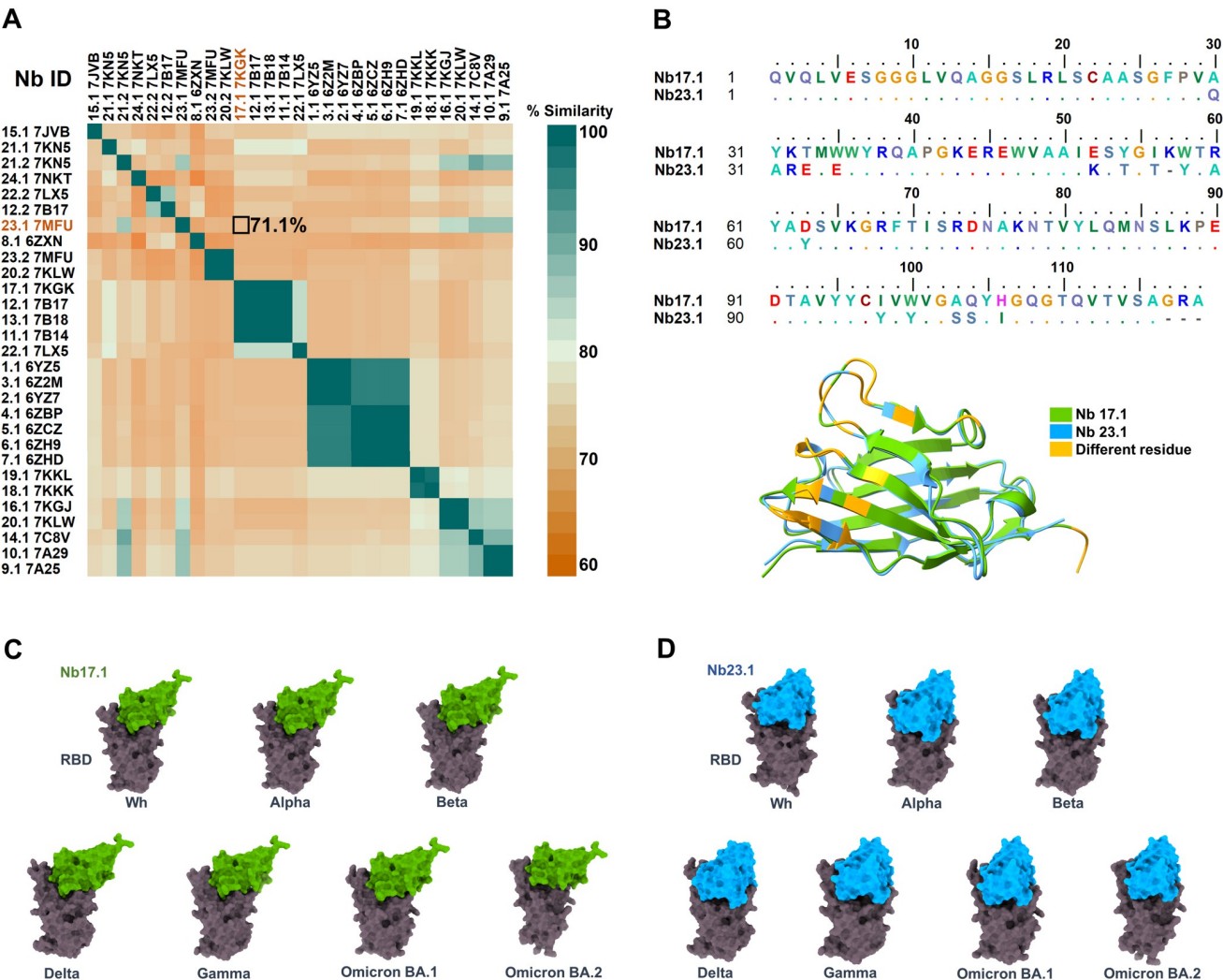

**Fig 3. Similarity of selected Nbs.** (A) Nb sequence similarity, (B) Sequence and structure alignment of two selected Nbs, Nb17.1 and Nb23.1, where the different residues on the binding domain are shaded by yellow and (C-D) Binding pose of the two lead Nbs on different targeted RBDs. Note that xx.1 refers to the Nb that interacts with the RBM of the RBD, while xx.2 interacts with other domains of the RBD.

Nb17.1 and Nb23.1 had different interaction postures on the RBM of all RBDs. Based on these results, Nb17.1 and Nb23.1 met all four selection criteria and were selected as lead Nbs for further modification to broad-specific Nbs for targeted RBDs.

## Engineering of Nbs for target RBDs

To improve the binding affinity and specificity of the two lead Nbs, Nb17.1 and Nb23.1, the modification of engineered Nbs centered on crucial target residues at the interaction site of Nbs to each target RBD, while considering the non-contact/interaction amino acid(s) and amino acid(s) that caused unfavorable interactions (S1 Fig). This study selected amino acid candidates based on their ability to contribute to hydrogen bonding and hydrophobic interactions with RBD, as described previously [40]. The single-point mutated lead Nbs and target RBDs were docked by the HDOCK program to determine the contribution of each mutated residue to the affinity and specificity to targeted RBDs.

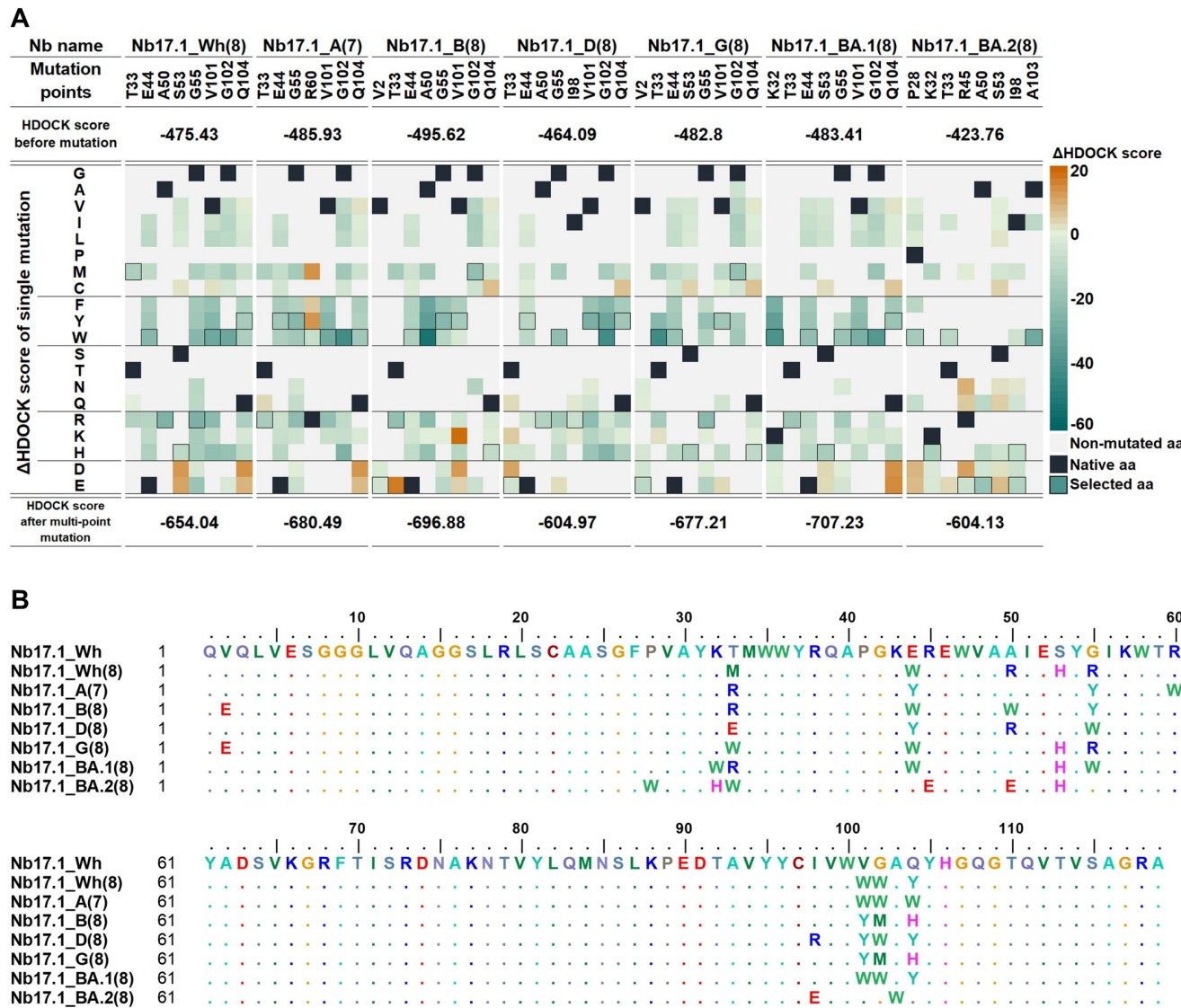

**Fig 4. The mutation point of Nb17.1.** (A) ΔHDOCK score and (B) sequence for each engineered Nb17.1.

In the introduction of single site-directed mutations (Figs 4A and 5A), each mutation site was altered with various amino acids possessing diverse properties. For instance, at the intriguing mutation sites, there are glycine (G), alanine (A), and valine (V) representing the category of small, non-polar amino acids, while serine, threonine (T), and glutamine (Q) are among the polar amino acids. Additionally, glutamic acid (E) and lysine (K) fall into the group of charged amino acids. These specific amino acids are prominently found at noteworthy mutation sites. To assess the contribution of each mutated amino acid towards affinity and specificity to the target RBDs, the single-point mutated lead Nbs and target RBD were subjected to docking by the HDOCK program to compute the difference in docking score (ΔHDOCK score) before and after mutation. The mutated residue with the lowest ΔHDOCK score at each mutation position of lead Nb specific to each target RBD was then selected. Interestingly, the majority of mutated residues of both Nb17.1 and Nb23.1 were tyrosine (Y)

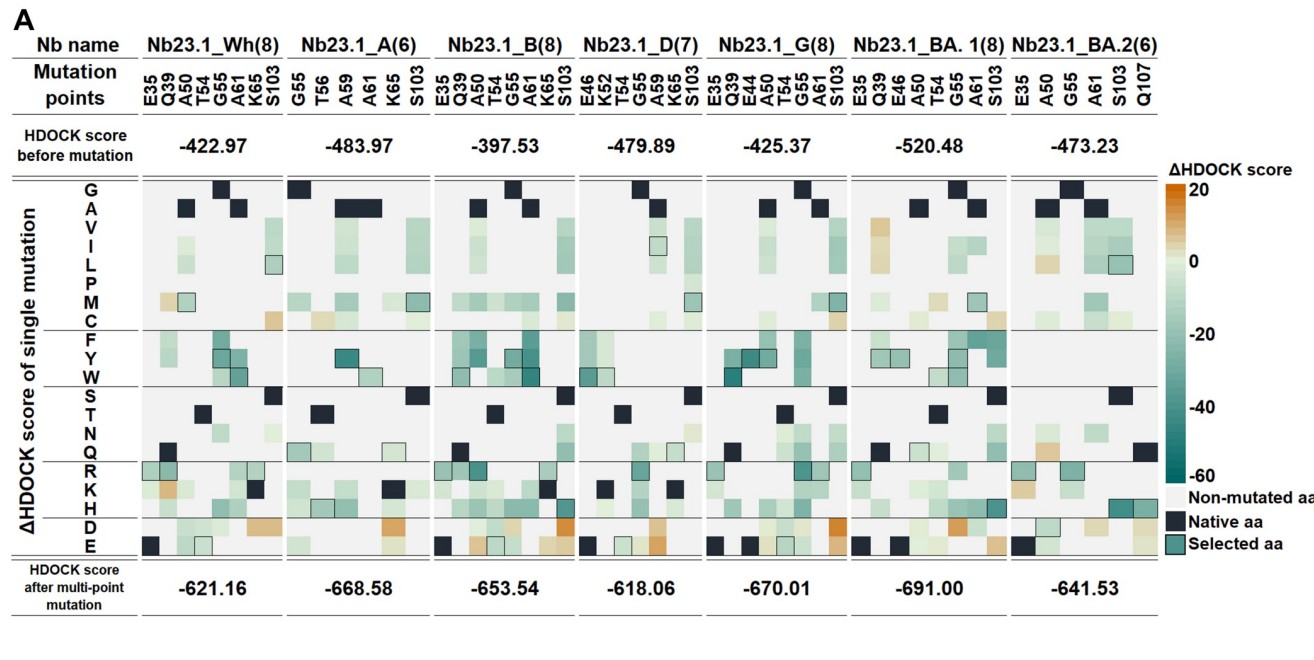

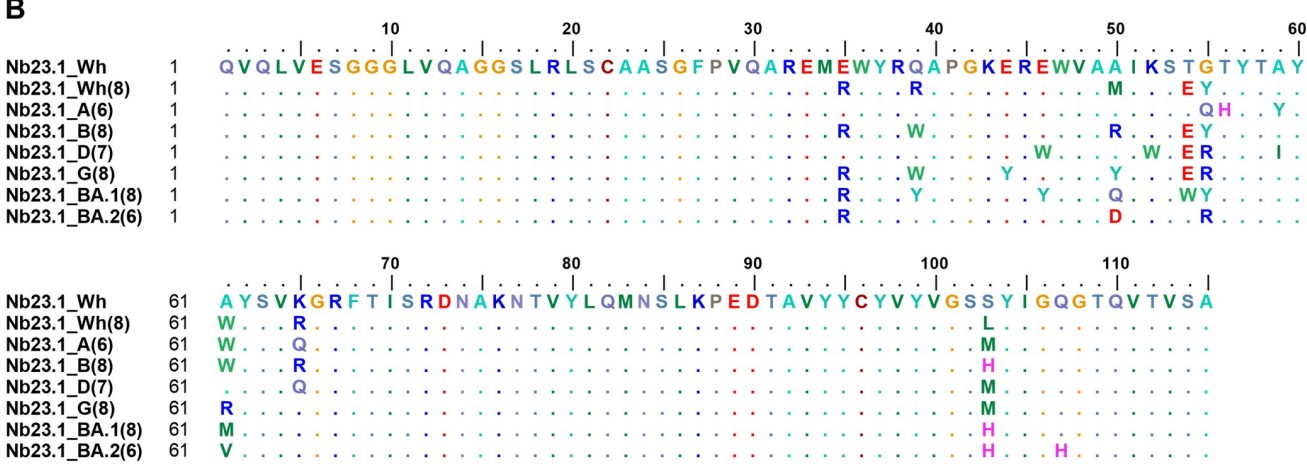

**Fig 5. The mutation point of Nb23.1.** (A) ΔHDOCK score and (B) sequence for each engineered Nb23.1.

and tryptophan (W), amino acids that contain an aromatic ring at the side chain (see sequence alignment in Figs 4B and 5B). The best mutated residues from docking study were combined for multi-point mutation to achieve better docking scores than a single-point mutation. As a result, seven engineered Nb17.1 and Nb23.1 (14 new Nbs in total) were obtained, which strongly interacted with their RBD. For example, Nb17.1_A(7), where "Nb17.1" was the original code of lead Nb, "A" was the Alpha variant of RBD, and "(7)" was the number of mutated residues. The docking score of the engineered Nb17.1 was Nb17.1_BA.1(8) (-707.23), Nb17.1_B(8) (-696.88), Nb17.1_A(7) (-680.49), Nb17.1_G(8) (-677.21), while the engineered Nb23.1 were Nb23.1_BA.1(8) (-691.00), Nb23.1_G(8) (-670.01), and Nb23.1_A(6) (-668.58).

The interaction investigation between engineered Nb17.1 (or Nb23.1) and targeted RBDs shown in Figs 6–8 Fig 7, Fig 8 indicates that multi-point mutation of the two lead Nbs resulted

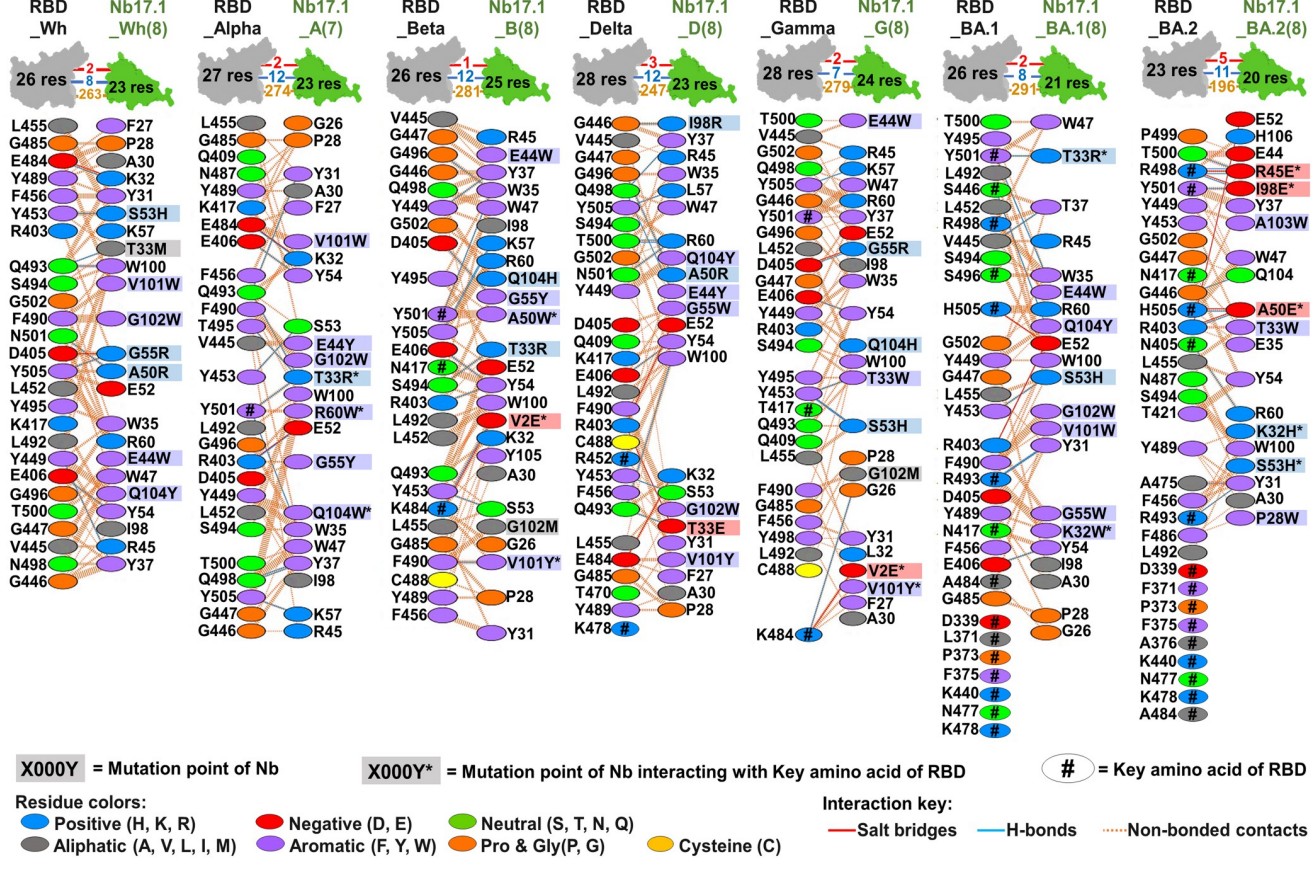

**Fig 6. The protein-protein interactions of the engineered Nb17.1 with various targeted RBDs.**

in improved docking scores compared to their native Nbs as shown in S7 and S8 Figs, as evidenced by more contacted residues, and increased chemical interactions such as salt bridges, hydrogen bonds, and non-bonded contacts. This improvement demonstrated an increase in specificity between the lead Nbs and RBD due to the multi-point mutation. As anticipated from the mutation, it was observed that certain altered amino acids in Nb could engage with the crucial amino acid in RBD. Additionally, the docking yields revealed that hydrophobic interactions played a critical role in Nb improvement, as they are residues found in all VOC RBDs. The hydrophobic complementary contact region on RBM (S9–S11 Figs) is rich in non-polar and aromatic amino acids, such as glycine, alanine, leucine, isoleucine, proline, valine, tyrosine, and phenylalanine [41, 42].

## Physicochemical properties prediction of Engineered Nbs

The physicochemical properties of the engineered Nbs, including molecular weight, extinction coefficient, aliphatic index, grand average of hydropathicity (GRAVY), instability index, theoretical pI, and solubility, were predicted to evaluate their biological activity in experimental assays and therapeutics (S4 and S5 Tables and S12 Fig). The aromatic mutated residues of engineered Nbs exhibited a higher extinction coefficient (see S4 and S5 Tables). The aliphatic-rich residues (glycine, alanine, and valine of engineered Nb17.1 and glycine and alanine of engineered Nb23.1) at the interaction sites tended to be mutated to aromatic residues such as G55Y

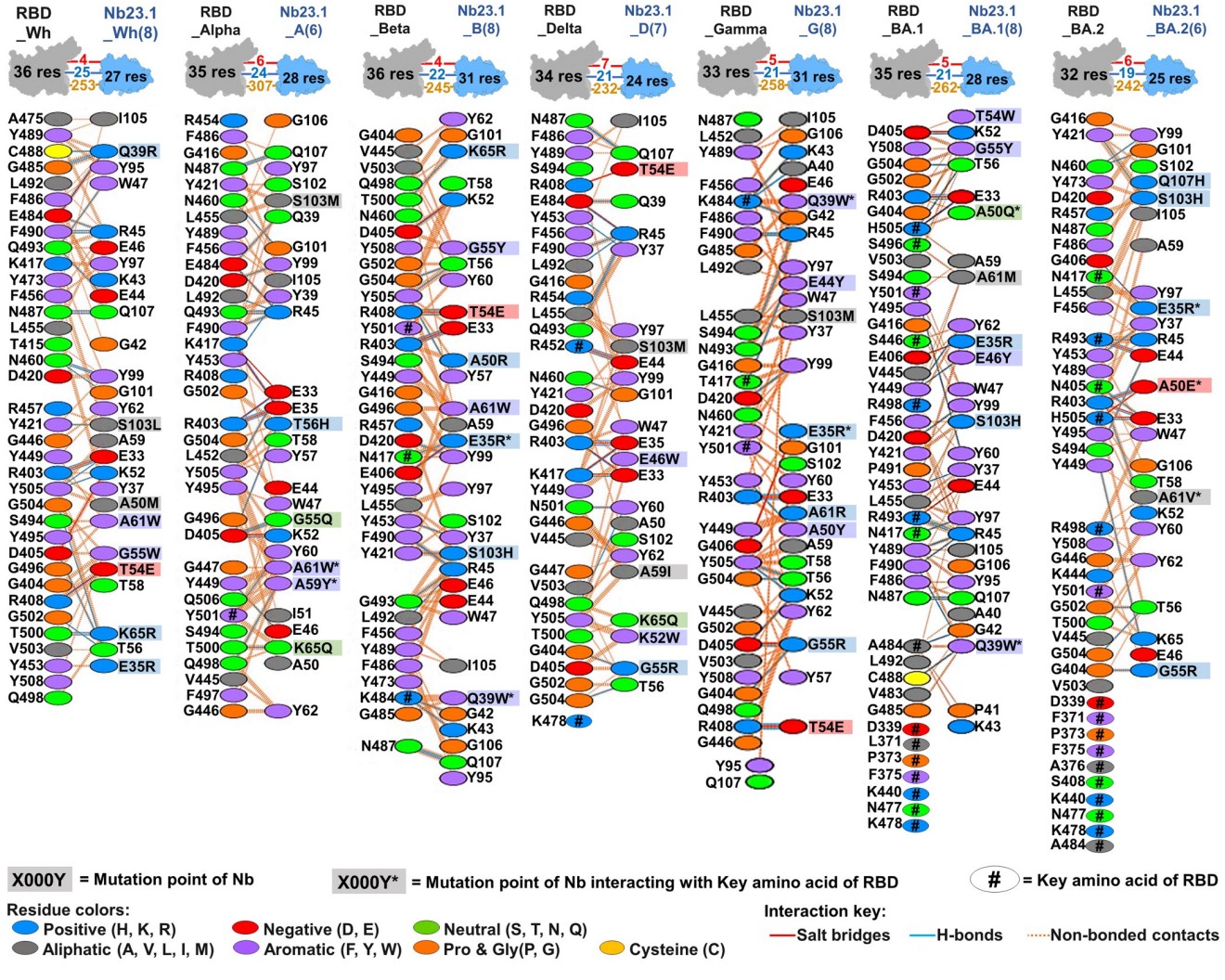

**Fig 7. The protein-protein interactions of the engineered Nb23.1 with various targeted RBDs.**

of Nb17.1_A(7), Nb17.1_B(8), and Nb23.1_B(8), resulting in a decrease in the aliphatic index value (S12A Fig) and an increase in hydrophobicity as indicated by the GRAVY value (S12B Fig). The mutated residues of Nb17.1 and Nb23.1 also affected the stability of all engineered Nbs, as shown by the increase in the instability index value (S12C Fig). The predicted solubility of engineered Nbs was mostly decreased from engineered Nb17.1, and some engineered Nb23.1 (S12D Fig), which could be attributed to the increase of aromatic and non-polar amino acid residues such as tryptophane and tyrosine on the engineered Nbs surface after mutation [43]. The theoretical pI value was a critical parameter for protein aggregation, as its decrease to the biological pH of 7.4 was observed in Nb17.1_BA.2(8), Nb23.1_A(6), and Nb23.1_D(7) compared to native Nb (S12E Fig). As well as the total charge of engineered Nbs was predicted, the almost engineered Nb17.1 revealed the decreasing total charge. Despite this, the engineered Nb23.1 tended to increase of total charge, except the Nb23.1_A(6) and Nb23.1_D(7) (S12F Fig). The calculated total charge is linked to other predicted physiochemical properties. In real experiments, decreased solubility and total charge may result in protein aggregation. To improve the solubility, pI, and stability of Nbs for biological research and applications, potential procedures

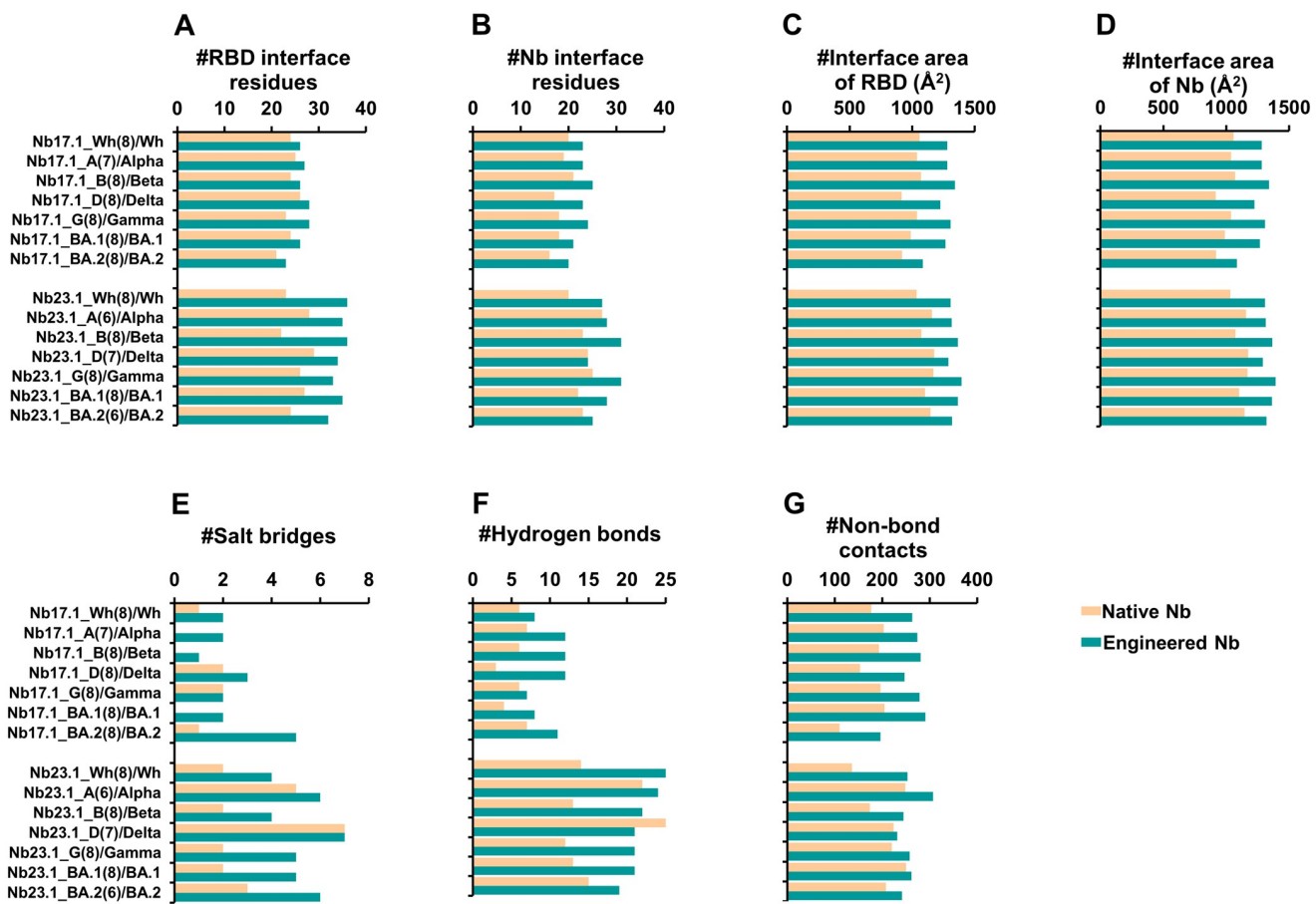

**Fig 8. The comparison of protein-protein interactions between native and engineered Nb17.1 and Nb23.1 with various targeted RBDs.** The following parameters were analyzed: (A) number of RBD interface residues, (B) number of Nb interface residues, (C) interface area of RBD, (D) interface area of Nb, (E) number of salt bridge interactions, (F) number of hydrogen bonds, and (G) number of non-bonded contacts.

include the fusion of polycationic amino acid tags to reduce protein aggregation and the incorporation of ligands to improve stability and bioavailability [44].

### Broad-specific evaluation of engineered Nbs

To investigate the broad specific binding of engineered Nbs, cross-docking of all 14 engineered Nbs to all target RBDs was performed, with native Nbs and ACE2 included for comparison. Cross-docking to other viral RBDs/HAs, including human coronaviruses (e.g., SARS-CoV-1, MERS, HCoV-229E, HCoV-NL63, HCoV-HKU1, and HCoV-OC43 [45–47]) and influenza viruses (e.g., H3N2 and H1N1 [48, 49]), was also performed to identify the specific binding of engineered Nbs, as these viruses share some similar symptoms and infection to SARS-CoV-2 [47, 48]. The docking results are shown in Fig 9. By comparing HDOCK scores of engineered Nbs to ACE2 or native Nbs, it was found that the binding affinity of all engineered Nb17.1 and Nb23.1 to targeted RBDs was meaningful. The docking score of Nb17.1_BA.2(8) was significantly better than ACE2, but not substantially different from the native Nb17.1. The evaluation of the specificity of engineered Nbs by cross-docking with other viral RBDs/HA showed that all engineered Nbs had a higher binding affinity to their target RBDs than other viral RBDs/Has. As expected, although all engineered Nb17.1 and Nb23.1 were multi-point mutations,

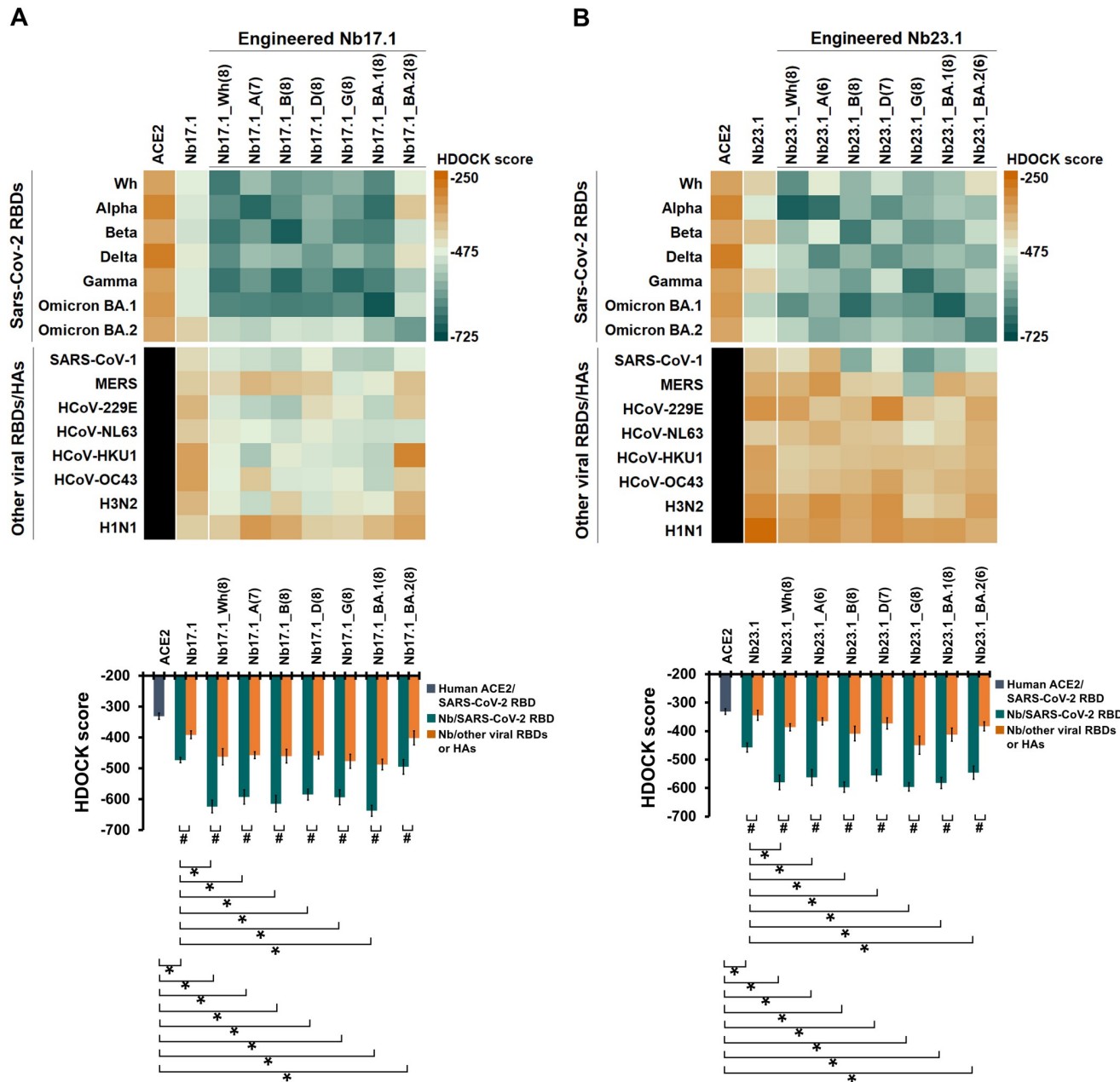

**Fig 9. Broad-specific evaluation of engineered Nbs.** The heat map of HDOCK score of the engineered Nbs (A) Nb17.1 and (B) Nb23.1 with different targeted RBDs and ACE2 receptor, where the mean ± SEM values are plotted below (N = 7 for target RBDs and N = 8 for other viral RBDs/HAs). Statistical analysis was performed using independent t-test (two-tailed), with significant differences denoted by "#" for $p < .05$, and One-Way ANOVA with the Dunnett test (two-tailed), with significant differences denoted by "*" for $p < .05$.

these engineered Nb can bind on the RBM motif as well as the native lead Nbs (S13 Fig). This result indicated that the engineered Nbs could be applied for SARS-CoV-2 broad-specific neutralization [7–9]. The finding represents an exciting outcome with a usable yield however, further investigation is required in a biological system to prove their effectiveness in a real environment for the next generation of usability as SARS-CoV-2 prevention, treatment, or diagnosis applications.

## Discussion

In this research, we have identified four notable findings regarding the Nb/RBD docking approach. Firstly, we determined that the protein-protein molecular docking technique utilizing HDOCK was suitable for evaluating the interaction between Nb/RBD affinity during Nb screening and engineering, enabling the achievement of broad specificity towards VOC RBDs [20, 50]. Secondly, as part of the lead Nb selection process, we identified two prominent Nbs with the lowest HDOCK scores, namely Nb17.1 and Nb23.1, out of the 29 Nbs evaluated. These Nbs displayed exceptional binding affinity and successfully engaged with the RBM of all RBDs. The third finding of this study involves the further engineering the lead Nbs to generate novel Nbs capable of effectively interacting with all VOC RBDs. A significant improvement in docking scores for Nb/RBD interactions was achieved by incorporating single-point mutations at specific residues on the Nb interface. Additionally, the multi-point engineered Nbs displayed exceptional binding affinity and demonstrated high specificity towards all VOC RBDs [51]. The presence of mutated aromatic or non-polar residues on the Nbs played a crucial role in achieving complementary binding to the VOC RBDs. This can be attributed to the hydrophobic nature of these residues, which facilitated favorable interactions with the corresponding hydrophobic regions on the RBD [41, 42]. The docking scores of the novel 14 engineered Nbs were higher than the native Nb, indicating an enhancement in their binding capabilities after the redesign process. Specific mutated residues showed efficient interactions with vital amino acids in the VOC RBDs, demonstrating a broad-specific interaction across all VOC RBDs. Final finding: as anticipated, among the engineered Nbs, Nb17.1_Wh(8) and Nb17.1_BA.1(8) exhibited the highest binding affinity towards all VOC RBDs, suggesting their potential for effectively inhibiting RBD_BA.1, the predominant VOC responsible for global transmission [5]. Moreover, the engineered Nbs Nb17.1_Wh(8) and Nb17.1_BA.1(8) demonstrated exceptional broad neutralization activity against all VOC RBDs [52], as evidenced by their low docking scores and minimal alteration in their physicochemical properties. Importantly, these engineered Nbs exhibited a high isoelectric point (pI) value and a positive charge, which can be attributed to the presence of arginine residues in their structure. This characteristic is commonly observed in nanobodies and contributes to their resistance against aggregation and maintaining stability, further enhancing their potential as therapeutic agents [53]. Consequently, the engineered Nbs show potential for the prevention and treatment of various SARS-CoV-2 variants. However, further experimental investigations are crucial to validate and select the most potent neutralizing Nbs for future applications.

## Supporting information

**S1 Table. The general information regarding the Nb/RBD and Ab/RBD datasets sourced from the Protein Data Bank (PDB).**
(PDF)

**S2 Table. The general information of targeted RBDs obtained from PDB.**
(PDF)

**S3 Table. The general information of ACE2 and other viral RBDs/HA obtained from PDB.**
(PDF)

**S4 Table. The physicochemical properties of native and engineered Nb17.1.**
(PDF)

**S5 Table. The physicochemical properties of native and engineered Nb23.1.**
(PDF)

**S1 Fig. An example of the consideration of single-point mutation residue.** (A) representing a non-interacting residue, (B) an unfavorable interaction, and (C-D) the protein-protein interaction after mutation.
(TIF)

**S2 Fig. Evaluation of the docking programs.** The RMSD distribution of (A) the top pose and (B) the best pose, where the asterisk indicates a significant difference (p < 0.05, Kruskal-Wallis with Dunn's test) between programs. (C) The success of docking programs.
(TIF)

**S3 Fig. The RMSD distribution of the best and top poses obtained by various docking methods.** (A) HDOCK, (B) ATTRACT, (C) pyDockWEB, (D) GRAMM-X, (E) PatchDock, (F) FRODOCK, and (G) ZDOCK, for Nb/RBD and Ab/RBD complexes. The statistical significance was evaluated using a paired-sample Wilcoxon signed-rank test (* denotes significance at p < 0.05, N = 115).
(TIF)

**S4 Fig. The impact of the number of top docking poses generated by different docking programs on the docking performance.** (A) the mean RMSD ± SEM and (B) success rate. Blind docking was performed on Nb/RBD and Ab/RBD datasets, with N = 115.
(TIF)

**S5 Fig. The impact of the RMSD criteria on the success rate.** (A) top pose and (B) best pose obtained by different docking methods for Nb/RBD and Ab/RBD complexes, with N = 115.
(TIF)

**S6 Fig. The impact of the RMSD criteria on the success rate of docking methods, with the number of complexes during blind docking presented for cases.** (A) the top pose RMSD and (B) the best pose RMSD were considered.
(TIF)

**S7 Fig. The protein-protein interactions of Nb17.1 with various targeted RBDs.**
(TIF)

**S8 Fig. The protein-protein interactions of Nb23.1 with various targeted RBDs.**
(TIF)

**S9 Fig. The sequences and structure of RBD.** (A) The sequence alignment of SARS-CoV-2 RBDs and the comparison of RBM sequence in yellow border, and (B) interaction of ACE2/SARS-CoV-2 RBD and Nb/SARS-CoV-2 RBD.
(TIF)

**S10 Fig. The structure and surface hydrophobicity of SARS-CoV-2 RBD, ACE2, and other viral RBDs/HAs.**
(TIF)

**S11 Fig. The 3D structure and mutation residue(s) of targeted RBDs.**
(TIF)

**S12 Fig. The physicochemical properties of engineered Nbs.** (A) aliphatic index, (B) grand average of hydropathicity (GRAVY), (C) instability index, (D) theoretical pI, (E) solubility, and (F) total charge of engineered (green bar) Nb17.1 and (blue bar) Nb23.1.
(TIF)

**S13 Fig. The docking pose of the engineered Nbs.** (A) Nb17.1 and (B) Nb23.1 with different targeted RBDs.
(TIF)

## Acknowledgments

P.L. would like to thank to Petchra Pra Jom Klao Ph.D. Research Scholarship from King Mongkut's University of Technology Thonburi. We extend our gratitude to Dr. Nitchakan Darai and Miss Hathaichanok Chuntakaruk (Program in Bioinformatics and Computational Biology, Graduate School, Chulalongkorn University) for their insightful discussions and technical assistance. We thank the ASEAN-European Academic University Network (ASEAUNINET) for a short visit grant. The Vienna Scientific Cluster (VSC) is acknowledged for facilities and computing resources.

## Author Contributions

**Conceptualization:** Thanyada Rungrotmongkol, Nongluk Plongthongkum, Kittikhun Wangkanont, Peter Wolschann, Rungtiva P. Poo-arporn.

**Data curation:** Phoomintara Longsompurana.

**Funding acquisition:** Thanyada Rungrotmongkol, Rungtiva P. Poo-arporn.

**Investigation:** Phoomintara Longsompurana, Thanyada Rungrotmongkol, Peter Wolschann, Rungtiva P. Poo-arporn.

**Methodology:** Phoomintara Longsompurana, Thanyada Rungrotmongkol, Nongluk Plongthongkum, Kittikhun Wangkanont, Peter Wolschann, Rungtiva P. Poo-arporn.

**Project administration:** Thanyada Rungrotmongkol, Rungtiva P. Poo-arporn.

**Resources:** Rungtiva P. Poo-arporn.

**Supervision:** Thanyada Rungrotmongkol, Rungtiva P. Poo-arporn.

**Validation:** Thanyada Rungrotmongkol.

**Visualization:** Phoomintara Longsompurana, Thanyada Rungrotmongkol.

**Writing – original draft:** Phoomintara Longsompurana.

**Writing – review & editing:** Thanyada Rungrotmongkol, Nongluk Plongthongkum, Kittikhun Wangkanont, Peter Wolschann, Rungtiva P. Poo-arporn.

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
