## [Decision Letter · Decision Letter 0]

25 Aug 2023

PONE-D-23-16784Computational design of novel nanobodies targeting the receptor binding domain of variants of concern of SARS-CoV-2PLOS ONE

Dear Dr. Poo-arporn,

Thank you for submitting your manuscript to PLOS ONE. After careful consideration, we feel that it has merit but does not fully meet PLOS ONE’s publication criteria as it currently stands. Therefore, we invite you to submit a revised version of the manuscript that addresses the points raised during the review process.

We look forward to receiving your revised manuscript.

Kind regards,

Sheikh Arslan Sehgal, PhD

Academic Editor

PLOS ONE

Journal Requirements:

"This research project is supported by the National Research Council of Thailand (NRCT) and King Mongkut’s University of Technology Thonburi: N42A650316, the Research Strengthening Project of the Faculty of Engineering and Thailand Science Research and Innovation (TSRI), Basic Research Fund: Fiscal year 2023 under project number FRB660073/0164  (Program Smart Healthcare). P.L. would like to thank the Petchra Pra Jom Klao Ph.D. Research Scholarship from King Mongkut’s University of Technology Thonburi. We extend our gratitude to Dr. Nitchakan Darai and Miss Hathaichanok Chuntakaruk (Program in Bioinformatics and Computational Biology, Graduate School, Chulalongkorn University) for their insightful discussions and technical assistance. T.R. is financially supported by the National Research Council of Thailand (NRCT, grant number N42A650231)."

"This research project is supported by the National Research Council of Thailand (NRCT) and King Mongkut’s University of Technology Thonburi: N42A650316, the Research Strengthening Project of the Faculty of Engineering and Thailand Science Research and Innovation (TSRI), Basic Research Fund: Fiscal year 2023 under project number FRB660073/0164  (Program Smart Healthcare). P.L. would like to thank the Petchra Pra Jom Klao Ph.D. Research Scholarship from King Mongkut’s University of Technology Thonburi. T.R. is financially supported by the National Research Council of Thailand (NRCT, grant number N42A650231). The funders had no role in study design, data collection and analysis, decision to publish, or preparation of the manuscript."

5. Please upload a new copy of all Figures as the detail is not clear. Please follow the link for more information: " ext-link-type="uri" xlink:type="simple">https://blogs.plos.org/plos/2019/06/looking-good-tips-for-creating-your-plos-figures-graphics/"
https://blogs.plos.org/plos/2019/06/looking-good-tips-for-creating-your-plos-figures-graphics/

Reviewers' comments:

Reviewer's Responses to Questions

**Comments to the Author**

1. Is the manuscript technically sound, and do the data support the conclusions?

Reviewer #1: Partly

2. Has the statistical analysis been performed appropriately and rigorously? 

Reviewer #1: N/A

3. Have the authors made all data underlying the findings in their manuscript fully available?

Reviewer #1: No

4. Is the manuscript presented in an intelligible fashion and written in standard English?

Reviewer #1: No

5. Review Comments to the Author

Reviewer #1: All main figures are missing and are not included in the manuscript file for review. Without the complete data/figures available, the reviewer cannot make a sound judgement on the quality of the manuscript.

6. PLOS authors have the option to publish the peer review history of their article (what does this mean?). If published, this will include your full peer review and any attached files.

Reviewer #1: No

---

## [Author Response · Author response to Decision Letter 0]

6 Sep 2023

We have responded to specific reviewer and editor comments, as outlined in the 'Response to Reviewers' document.

---

## [Decision Letter · Decision Letter 1]

25 Sep 2023

PONE-D-23-16784R1Computational design of novel nanobodies targeting the receptor binding domain of variants of concern of SARS-CoV-2PLOS ONE

Dear Dr. Poo-arporn,

Thank you for submitting your manuscript to PLOS ONE. After careful consideration, we feel that it has merit but does not fully meet PLOS ONE’s publication criteria as it currently stands. Therefore, we invite you to submit a revised version of the manuscript that addresses the points raised during the review process.

We look forward to receiving your revised manuscript.

Kind regards,

Sheikh Arslan Sehgal, PhD

Academic Editor

PLOS ONE

Journal Requirements:

Additional Editor Comments:

Figures are very poor and have poor resolution.

Reviewers' comments:

Reviewer's Responses to Questions

**Comments to the Author**

1. If the authors have adequately addressed your comments raised in a previous round of review and you feel that this manuscript is now acceptable for publication, you may indicate that here to bypass the “Comments to the Author” section, enter your conflict of interest statement in the “Confidential to Editor” section, and submit your "Accept" recommendation.

Reviewer #1: All comments have been addressed

2. Is the manuscript technically sound, and do the data support the conclusions?

Reviewer #1: Yes

3. Has the statistical analysis been performed appropriately and rigorously? 

Reviewer #1: Yes

4. Have the authors made all data underlying the findings in their manuscript fully available?

Reviewer #1: Yes

5. Is the manuscript presented in an intelligible fashion and written in standard English?

Reviewer #1: Yes

6. Review Comments to the Author

Reviewer #1: The authors have addressed all the comments. There is a grammatical error in Line 221. Change to "there are glycine (G), ... ".

7. PLOS authors have the option to publish the peer review history of their article (what does this mean?). If published, this will include your full peer review and any attached files.

Reviewer #1: No

---

## [Author Response · Author response to Decision Letter 1]

5 Oct 2023

We have addressed your comments in the attached file titled 'Response to Reviewers.

---

## [Editor Report · Decision Letter 2]

10 Oct 2023

Computational design of novel nanobodies targeting the receptor binding domain of variants of concern of SARS-CoV-2

PONE-D-23-16784R2

Dear Dr. Poo-arporn,

We’re pleased to inform you that your manuscript has been judged scientifically suitable for publication and will be formally accepted for publication once it meets all outstanding technical requirements.

Kind regards,

Sheikh Arslan Sehgal, PhD

Academic Editor

PLOS ONE
---

## [Editor Report · Acceptance letter]

16 Oct 2023

PONE-D-23-16784R2 

 Computational design of novel nanobodies targeting the receptor binding domain of variants of concern of SARS-CoV-2 

Dear Dr. Poo-arporn:

I'm pleased to inform you that your manuscript has been deemed suitable for publication in PLOS ONE. Congratulations! Your manuscript is now with our production department. 

Kind regards, 

on behalf of

Dr Sheikh Arslan Sehgal 

Academic Editor

PLOS ONE